# 3D-GRES: Generalized 3D Referring Expression Segmentation

Changli Wu*
wuchangli@stu.xmu.edu.cn
Key Laboratory of Multimedia
Trusted Perception and Efficient
Computing,
Ministry of Education of China,
Xiamen University
Xiamen, Fujian, China

Yihang Liu*
liuyihang@stu.xmu.edu.cn
Key Laboratory of Multimedia
Trusted Perception and Efficient
Computing,
Ministry of Education of China,
Xiamen University
Xiamen, Fujian, China

Jiayi Ji
jjyxmu@gmail.com
Key Laboratory of Multimedia
Trusted Perception and Efficient
Computing,
Ministry of Education of China,
Xiamen University
Xiamen, Fujian, China

Yiwei Ma
yiweima@stu.xmu.edu.cn
Key Laboratory of Multimedia
Trusted Perception and Efficient
Computing,
Ministry of Education of China,
Xiamen University
Xiamen, Fujian, China

Haowei Wang
asucawang@tencent.com
Youtu Lab, Tencent,
Shanghai, China

Gen Luo
luogen@stu.xmu.edu.cn
Key Laboratory of Multimedia
Trusted Perception and Efficient
Computing,
Ministry of Education of China,
Xiamen University
Xiamen, Fujian, China

Henghui Ding
henghui.ding@gmail.com
Institute of Big Data,
Fudan University,
Shanghai, China

Xiaoshuai Sun
xssun@xmu.edu.cn
Key Laboratory of Multimedia
Trusted Perception and Efficient
Computing,
Ministry of Education of China,
Xiamen University
Xiamen, Fujian, China

Rongrong Ji†
rrji@xmu.edu.cn
Key Laboratory of Multimedia
Trusted Perception and Efficient
Computing,
Ministry of Education of China,
Xiamen University
Xiamen, Fujian, China

## Abstract

3D Referring Expression Segmentation (3D-RES) is dedicated to segmenting a specific instance within a 3D space based on a natural language description. However, current approaches are limited to segmenting a single target, restricting the versatility of the task. To overcome this limitation, we introduce Generalized 3D Referring Expression Segmentation (3D-GRES), which extends the capability to segment any number of instances based on natural language instructions. In addressing this broader task, we propose the Multi-Query Decoupled Interaction Network (MDIN), designed to break down multi-object segmentation tasks into simpler, individual segmentations. MDIN comprises two fundamental components: Text-driven Sparse Queries (TSQ) and Multi-object Decoupling Optimization (MDO). TSQ generates sparse point cloud features distributed over key targets as the initialization for queries. Meanwhile, MDO is tasked with assigning each target in multi-object scenarios to different queries while maintaining their semantic consistency. To adapt to this new task, we build a new dataset, namely Multi3DRes. Our comprehensive evaluations on this dataset demonstrate substantial enhancements over existing models, thus charting a new path for intricate multi-object 3D scene comprehension. The benchmark and code are available at https://github.com/sosppxo/MDIN.

## CCS Concepts

• **Computing methodologies** → **Scene understanding**; **Computer vision tasks**.

## Keywords

Generalized 3D Referring Expression Segmentation, Query-based Mask Generation, Multimodal Contrastive Learning

*Equal contribution.
†Corresponding author.

*MM '24, October 28-November 1, 2024, Melbourne, VIC, Australia*
© 2024 Copyright held by the owner/author(s). Publication rights licensed to ACM.
ACM ISBN 979-8-4007-0686-8/24/10
https://doi.org/10.1145/3664647.3680841

## ACM Reference Format:

Changli Wu, Yihang Liu, Jiayi Ji, Yiwei Ma, Haowei Wang, Gen Luo, Henghui Ding, Xiaoshuai Sun, and Rongrong Ji. 2024. 3D-GRES: Generalized 3D Referring Expression Segmentation. In *Proceedings of the 32nd ACM International Conference on Multimedia (MM '24), October 28-November 1, 2024, Melbourne, VIC, Australia.* ACM, New York, NY, USA, 10 pages. https://doi.org/10.1145/3664647.3680841

# 1 Introduction

3D Referring Expression Segmentation (3D-RES) is a burgeoning direction in the multimodal domain, attracting widespread interest from researchers [26, 65]. This task aims to segment target instances based on given natural language expressions, distinguishing itself from 3D Referring Expression Comprehension (3D-REC) [1, 3, 4, 14, 16, 46, 81, 83], which merely locates objects with bounding boxes. 3D-RES is critical for applications in autonomous robotics, human-machine interaction, and self-driving systems, demanding not only object identification but also the generation of precise 3D masks.

However, traditional 3D Referring Expression Segmentation (3D-RES) settings [26, 34, 55, 65] are constrained to addressing single target cases, as depicted in Fig. 1-(1), a limitation significantly narrowing their practical application. In real-world scenarios, instructions often lead to situations where either no target is found or multiple targets need to be identified simultaneously, as shown in Fig. 1 (2)-(5). This reality presents a challenge that existing 3D-RES models are ill-equipped to handle. To bridge this gap, we introduce a new setting, Generalized 3D Referring Expression Segmentation (3D-GRES), designed to interpret instructions specifying an arbitrary number of targets. By enhancing the Multi3DRefer [80] dataset through the substitution of bounding boxes with masks, we develop the Multi3DRes dataset, crafted specifically for training and validation of 3D-GRES models.

The central challenge of 3D-GRES lies in accurately identifying multiple targets from a group of similar objects. For instance, distinguishing a specific table placed in a corner from numerous similar tables, as illustrated in Fig. 1-(4). Directly applying existing 3D-RES frameworks, such as 3D-STMN [65], to 3D-GRES tasks has proven to be ineffective. These frameworks typically employ a single query to activate the point cloud scene, culminating in a final mask. However, a single query struggles to precisely identify multiple targets among similar instances due to their identical appearances and similar semantics. Previous work [24, 35, 67, 70, 82] of 2D-GRES has progressively explored this area, providing a wealth of insights. Inspired by this, we recognize that the key to addressing such challenges lies in decoupling the task, allowing several queries to simultaneously handle the localization of a multi-object language instruction, with each query responsible for an individual instance within the multi-object scenario. However, their decoupling methods, such as Minimap [35], are designed for continuous 2D images and do not extend effectively to the unordered and sparse nature of 3D point clouds.

In this paper, we introduce the Multi-Query Decoupled Interaction Network (MDIN), a novel framework specifically engineered for 3D-GRES, which is responsible for facilitating the interaction between queries and both superpoints and text. To adeptly handle an arbitrary number of targets, we incorporate a mechanism that allows multiple queries to decouple and collaboratively generate multi-object results, with each query responsible for a single target within the multi-object instance, and a classification head that determines the query-wise presence of targets. To ensure that queries evenly cover key targets in the point cloud scene, we introduce a novel Text-driven Sparse Queries (TSQ) module to generate sparse, text-related queries. Furthermore, to simultaneously achieve

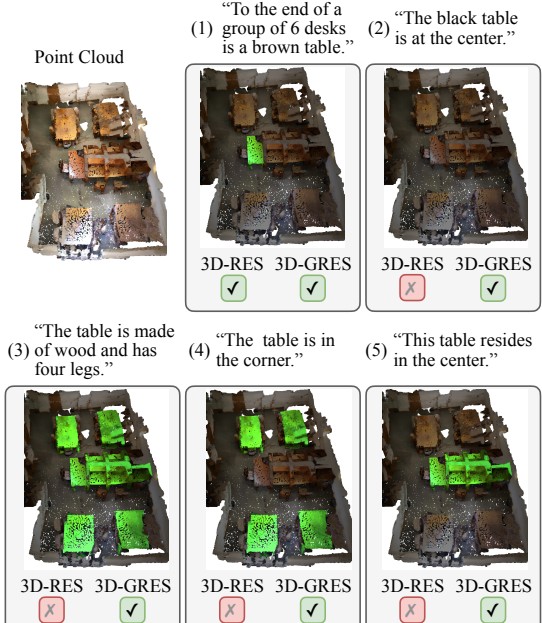

**Figure 1: Traditional 3D-RES is limited to single-target cases (1). In contrast, 3D-GRES can handle scenarios with any number of targets, including no target (2), single target, and multiple targets (3-5).**

distinctiveness among queries and maintain overall semantic coherence, we have developed a Multi-object Decoupling Optimization (MDO) strategy. This strategy decouples the multi-object mask into individual single-object supervisions, preserving the discriminative ability of each query. By anchoring the features of the queries and the superpoint features of the ground truth in the point cloud scene to the textual semantics, it ensures semantic consistency across multiple targets.

Our contributions are threefold:

- We introduce a new challenging 3D-GRES task and benchmark, exploring more general and flexible interaction in 3D scenes through the development of the MDIN.
- We design the TSQ to ensure balanced coverage of key targets within the point cloud scene and the MDO strategy to maintain the distinctiveness of queries while preserving the semantic consistency of instructions.
- Extensive quantitative and qualitative experiments demonstrate the effectiveness of our proposed methods in addressing 3D-GRES.

## 2 Related Work

### 2.1 2D Referring Expression Comprehension and Segmentation

Research in the multimodal field [11–13, 47–49, 68] is thriving due to its significant application value, with REC and RES tasks

receiving considerable attention. The 2D-REC task aims to predict a bounding box for the target based on a referring expression [8, 21, 23, 37, 39, 41, 53, 59, 76, 78, 85], while 2D-RES focuses on predicting a segmentation mask that accurately represents the referred object [22, 33, 36, 40, 44, 45, 72, 74, 75]. Common datasets for these tasks include ReferIt [30], RefCOCO [77], and RefCOCOg [50], where each expression refers to a single instance.

REC and RES methods are generally categorized into one-stage [9, 10, 25, 28, 29, 56, 63, 73] and two-stage [20, 38, 62, 76] approaches. One-stage methods fuse visual and linguistic features to directly predict segmentation masks, while two-stage methods generate instance proposals using object detection or segmentation and then use language features to select the target instances. Despite their success in the 2D domain, these methods face challenges in the 3D domain due to the irregularity and sparsity of 3D point clouds.

## 2.2 3D Referring Expression Comprehension and Segmentation

With the widespread adoption of deep learning techniques in 3D point clouds, the 3D-REC task has attracted significant attention. Chen et al. [3] released the ScanRefer dataset for the 3D visual grounding task, which involves locating referred objects in 3D scenes. Referit3d [1] proposes two datasets, Nr3d and Sr3d, where referred objects belong to fine-grained object categories, and scenes contain multiple instances of such categories. While most existing methods adopt a two-stage [4, 14, 16, 79, 83] paradigm, some researchers have also explored one-stage [46, 64] approaches.

In contrast, research on 3D-RES [17, 18, 26, 65] is still in its nascent stage. TGNN [26], as a pioneer in the 3D-RES task, proposed a two-stage method based on Graph Neural Networks. 3D-STMN [65] introduced a one-stage approach, significantly improving inference speed. These methods mainly focus on single-target descriptions, whereas our work is primarily dedicated to segmenting a flexible number of target objects based on language descriptions.

## 2.3 Generalized Referring Expression Comprehension and Segmentation

The original 2D-REC and RES tasks do not specify a limit on the number of target instances. However, previous datasets [30, 50, 77] typically assume each expression refers to a single target, which creates issues when no or multiple targets are present. To address this, Liu et al. [35] introduced the Generalized Referring Expression Segmentation (GRES) benchmark to handle expressions referring to zero, single, or multiple targets. He et al. [19] further developed this approach with Generalized Referring Expression Comprehension (GREC). Subsequent research [7, 24, 27, 67, 71] has expanded 2D-REC and RES tasks to include multiple target descriptions, with methods [70, 82, 84] showing strong performance on GRES.

In the 3D domain, research is limited. Zhang et al. [80] extended the ScanRefer task to Multi3DRefer, which locates a variable number of targets in 3D scenes using bounding boxes. However, bounding boxes can be ambiguous in dense scenes. We propose the 3D Generalized Referring Expression Segmentation (3D-GRES) task to improve localization by generating segmentation masks based on language descriptions.

## 3 3D-GRES

### 3.1 Classical 3D-RES

The classic 3D-RES task [26, 65] is focused on generating a 3D mask for a single target object within a point cloud scene, guided by a referring expression. This traditional task exhibits significant limitations. Firstly, it fails to accommodate scenarios where no object within the point cloud scene matches the given expression. An illustrative example of this is depicted in Fig. 1 (2), where no object corresponds to the expression "the black table is at the center". Secondly, it does not account for instances where multiple objects fit the described criteria. For example, in Fig. 1 (4), the expression "the table is in the corner" applies to four distinct objects. This significant gap between model capabilities and real-world applicability restricts the practical deployment of 3D-RES technologies in scenarios reflective of everyday complexities.

### 3.2 3D-GRES Settings

**Settings and Benchmark**. To overcome existing limitations, we introduce the Generalized 3D Referring Expression Segmentation (3D-GRES) task, designed to identify an arbitrary number of objects from textual descriptions. 3D-GRES involves processing a 3D point cloud scene $P$, a referring textual expression $E$, and generating a corresponding 3D mask $M$, which can signify zero, one, or multiple objects. For expressions indicating no targets, $M$ will be an all-zero mask, showing no selected points in $P$. It enables locating multiple objects through multi-target expressions and verifying the existence of specific objects in a scene with "nothing" expressions, thus offering enhanced flexibility and robustness in object retrieval and interaction within 3D environments.

**Multi3DRes Dataset**. To adapt to the 3D-GRES task, we utilize the Multi3DRefer dataset [80] to create the Multi3DRes dataset. The original Multi3DRefer dataset includes 61,926 language expressions referring to 11,609 objects across 800 ScanNet [6] scenes, with 6,688 expressions matching zero targets and 13,178 matching multiple targets. However, Multi3DRefer was designed for referring expression comprehension, yielding bounding boxes. To enhance precision, we construct the Multi3DRes dataset using instance masks from ScanNet [6]. This dataset extends the original by incorporating samples with no targets and multiple targets, thereby increasing task complexity and enabling more robust handling of real-world free-form user inputs.

**Metrics**. As for evaluation metrics, we employ the conventional mIoU (Mean Intersection over Union) and introduce Acc@$k$IoU, which measures the proportion of predicted masks with an IoU greater than $k$ compared to the ground truth masks, with $k$ belonging to the set {0.25, 0.5}. In cases where no target objects are present, we assume an IoU of 1 for correct predictions and an IoU of 0 for incorrect predictions. Specifically, for our proposed query decoupling method, MDIN, we adjust the IoU values of individual samples based on the predicted query-wise confidence $A^{tgt}$ indicating the presence of a target. If all queries are correctly predicted as zero-target (*i.e.,* $A^{tgt} > 0.5$ is a zero vector), their IoU is set to 1; otherwise, it is set to 0.

Furthermore, we categorize the samples into five classes based on the Multi3DRefer [80] manner: a) zero target without distractors of the same semantic class; b) zero target with distractors; c)

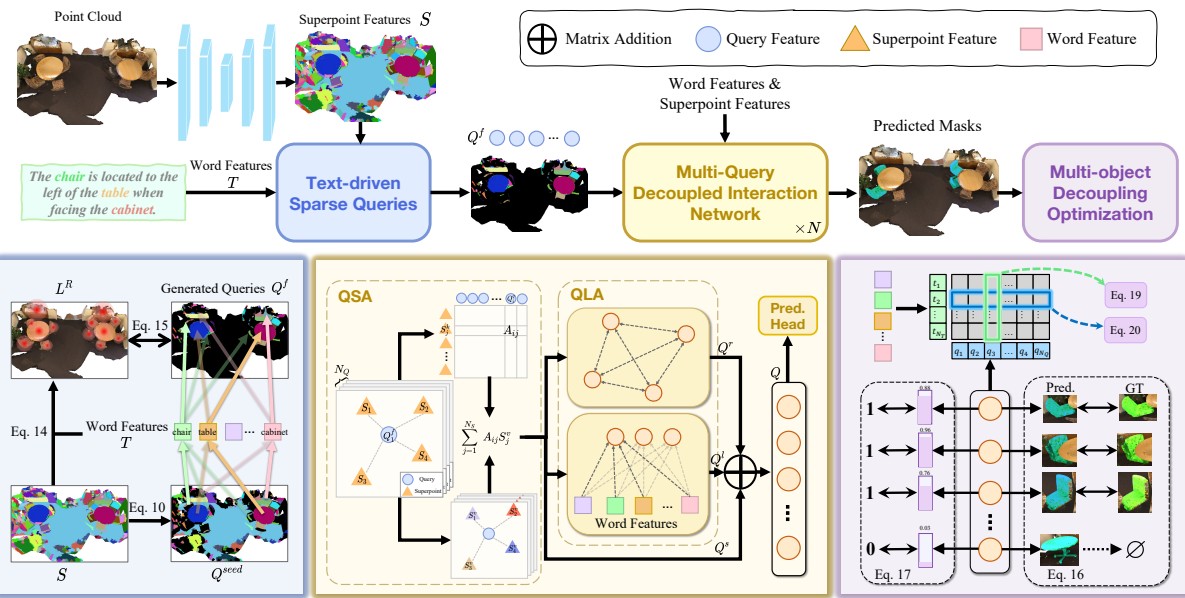

**Figure 2: The overall framework of MDIN, comprising its core modules TSQ and MDO. The input point cloud and text undergo feature extraction before being fed into the TSQ module to extract sparse decoupled queries. Subsequently, the MDIN module performs multimodal fusion and prediction. Finally, the MDO module carries out decoupled optimization.**

single target without distractors; d) single target with distractors; and e) multiple targets. Notably, classes c) and d) align with the "unique" and "multiple" cases, respectively, as defined in ScanRefer [3]. Classes a) and c) represent easier scenarios where either zero or a single target object of its semantic class is present in the scene, while classes b) and d) represent more challenging situations involving one or multiple target objects of the same semantic class in the scene.

## 4 Method

### 4.1 Feature Extraction

*4.1.1 Linguistic Feature and Text Decoupling.* In our approach, we first process the input referring expressions by encoding them into text tokens $\mathcal{T} \in \mathbb{R}^{N_T \times D_T}$, utilizing the pre-trained RoBERTa [42]. Here, $N_T$ represents the total number of tokens, and $D_T$ indicates the dimensionality of the original text features. To facilitate multimodal alignment, these encoded features are then mapped to a multimodal space of dimension $D$. This mapping is achieved through a linear transformation, denoted as $T = \mathcal{T} W_T$, where $T \in \mathbb{R}^{N_T \times D}$ represents the projected linguistic features, and $W_T \in \mathbb{R}^{D_T \times D}$ is a learnable parameter matrix.

Following Wu et al. [69], we use off-the-shelf natural language processing tools [57, 66] to parse descriptions into five semantic components: the Main object (the primary subject), the Auxiliary object (which helps locate the main object), Attributes (describing appearance and shape), Pronoun (substituting for the main object), and Relationship (the spatial relation between main and auxiliary objects). This breakdown facilitates a deeper understanding of entity relationships.

Next, we create position labels $L_{main}, L_{attri}, L_{auxi}, L_{pron}$, and $L_{rel} \in \mathbb{R}^{1 \times N_T}$ for words associated with each component, where $N_T$ represents the maximum text length, consistent with [69]. In these labels, the position of words about each component is marked with a 1, while all other positions are set to 0. To obtain the text feature of a decoupled component $t \in \mathbb{R}^{1 \times D}$, we perform a dot multiplication of the position label $L \in \mathbb{R}^{1 \times N_T}$ with the feature matrix of all words $T \in \mathbb{R}^{N_T \times D}$:

$$t = L \cdot T, \tag{1}$$

where $t_{main} = L_{main} \cdot T$, $t_{attri} = L_{attri} \cdot T$ and $t_{pron} = L_{pron} \cdot T$ are selected to form the positive word features $T^+$ for the target.

*4.1.2 Visual Feature and Superpoint Pooling.* For the input point cloud with positions $P^P \in \mathbb{R}^{N_P \times 3}$ and RGB $C_P \in \mathbb{R}^{N_P \times 3}$, the point-wise features $F^{point} \in \mathbb{R}^{N_P \times D_P}$ can be extracted through a Sparse 3D U-Net [15], where $N_P$ denotes the number of points and $D_P$ represents the original visual feature dimension. And we project $F^{point}$ into $D$-dimensional multimodal space: $F^P = F^{point} W_P$, where $F^P \in \mathbb{R}^{N_P \times D}$ is projected point feature and $W_P \in \mathbb{R}^{D_P \times D}$ is a learnable parameter. Then, we follow Sun et al. [60], Wu et al. [65] to obtain $N_S$ superpoints $\{K_i\}_{i=1}^{N_S}$ [32] from the original point cloud. Finally, we directly feed point-wise features $F^P$ into superpoint pooling layer [60] based on $\{K_i\}_{i=1}^{N_S}$ to get the superpoint-level features $S \in \mathbb{R}^{N_S \times D}$.

### 4.2 Multi-Query Decoupled Interaction Network

Prior work like 3D-STMN [65] used a single query to activate a mask on a point cloud scene, effectively localizing a single target in

3D-RES tasks. However, this approach struggles with multiple or unspecified targets. To address this, we introduce the Multi-Query Decoupled Interaction Network (MDIN) for 3D-GRES, inspired by Liu et al. [35]. MDIN uses multiple queries to handle individual instances in multi-target scenarios, aggregating these into a final result. For scenes without defined targets, predictions are made based on the confidence scores of each query, with zero targets predicted if all queries score low.

MDIN comprises multiple identical modules in series, each consisting of Query-Superpoint Aggregation (QSA) and Query-Language Aggregation (QLA) modules, facilitating interactions between queries, superpoints, and text. Unlike existing models like DETR [2] that use random initialization, MDIN employs a Text-driven Sparse Queries (TSQ) module to generate sparse queries $Q^f \in \mathbb{R}^{N_Q \times D}$ from $S$, ensuring effective scene coverage, as detailed in Sec. 4.3. To support multiple queries, we implement a Multi-object Decoupling Optimization (MDO) strategy to refine performance, detailed in Sec. 4.4.

*4.2.1 Query-Superpoint Aggregation (QSA).* The query $Q^f$ can be considered as an anchor in the point cloud scene [58]. By enabling queries to interact with superpoints, queries can capture the global information of a point cloud scene. Notably, the sampled superpoints serve as queries during this interaction process, allowing for stronger local aggregation. This local focus creates favorable conditions for the decoupling of queries. The architecture is depicted in Fig. 2. Initially, the similarity distribution is computed between the superpoint feature $S$ and query embeddings $Q^f$:

$$Q^q = Q^f W_{sq}, \quad S^k = SW_{sk},$$
$$A_{ij} = \frac{\text{Sim}\left(Q_i^q, S_j^k\right)}{\sum_{j=1}^{N_S} \text{Sim}\left(Q_i^q, S_j^k\right)}, \tag{2}$$

where $W_{sq}$ and $W_{sk}$ are learnable $D \times D$ parameters and $\text{Sim}(\cdot, \cdot)$ represents the similarity function, which in this case is defined as $\text{Sim}(q, k) = \exp(q \cdot k^T / \sqrt{D})$. Subsequently, the queries will aggregate their respective related superpoints using these similarity distributions:

$$S^v = SW_{sv},$$
$$Q_i^s = \sum_{j=1}^{N_S} A_{ij} S_j^v, \tag{3}$$

where $S \in \mathbb{R}^{N_S \times D}$ denotes the features of the superpoints, $W_{sv} \in \mathbb{R}^{D \times D}$ represents learnable parameters. Subsequently, the updated scene-aware $Q^s$ is fed into QLA for additional language-aware interactions.

*4.2.2 Query-Language Aggregation (QLA).* The scene-aware queries $Q^s$ come from collating superpoint features that do not contain relationships between queries and language information. We propose QLA module to model the query-query and query-language interactions. As in Fig. 2, QLA consists of a self-attention[61] for query features $Q^s$, and a multi-modal cross attention. The self-attention models the query-query dependency relationships. It computes the attention matrix by interacting one query with all other queries

and outputs the relation-aware query feature $Q^r$:

$$Q^r = \text{Softmax}(Q^s W_{qq}(Q^s W_{qk})^T) \cdot Q^s W_{qv}, \tag{4}$$

where $W_{qq}, W_{qk}, W_{qv} \in \mathbb{R}^{D \times D}$ are learnable parameters. Meanwhile, the query-language interactions follow the cross attention way, and firstly models the relationship between each word and each query:

$$A^l = \text{Softmax}(Q^s W_{lq}(TW_{lk})^T), \tag{5}$$

where $A^l \in \mathbb{R}^{N_Q \times N_T}$ and $W_{lq}, W_{lk} \in \mathbb{R}^{D \times D}$ are learnable parameters. Then it forms the language-aware query features using the derived word-query attention:

$$Q^l = A^l T. \tag{6}$$

Finally, the relation-aware query feature $Q^r$, language-aware query features $Q^l$, and scene-aware query features $Q^s$ are added together fused by an MLP:

$$Q = \text{MLP}(Q^s + Q^r + Q^l). \tag{7}$$

*4.2.3 Prediction Head.* For the $n$-th query, we first get its corresponding mask $M_n \in \{0, 1\}^{N_S}$ by multiplying its feature $Q_n$ with the mask branch of superpint features $S_M$. Meanwhile, $Q_n$ is also fed into an MLP to predict a confidence $A_n^{tgt}$ that indicates its probability of containing targets. We get the final predicted mask $\mathcal{M}$ by logically aggregating these masks:

$$S_M = SW_M, \quad M_n = \text{Binarization}(S_M \cdot Q_n),$$
$$\mathcal{M} = \text{Binarization}\left(\sum_n \mathbb{I}(A_n^{tgt} > 0.5) \cdot M_n\right), \tag{8}$$

where $S \in \mathbb{R}^{N_S \times D}$ is the superpoint features, $W_M \in \mathbb{R}^{D \times D}$ is learnable parameters, $\text{Binarization}(\cdot)$ denotes the binarization operation, which returns 1 when the input is a positive value and 0 otherwise, and $\mathbb{I}(A_n^{tgt} > 0.5)$ denotes whether the $n$-th query contains one of the target instances, which can be easily supervised by whether the query belongs to a particular GroundTruth instance due to its visual-based generation process.

## 4.3 Text-driven Sparse Queries

Directly using text tokens as queries [65] is suboptimal due to the entanglement of semantic descriptions in the text, making it difficult to differentiate between multiple objects. On the other hand, employing text-agnostic learnable parameters as queries [31, 60] requires a larger quantity to cover the space adequately and relies on unstable Hungarian matching, leading to difficulties in convergence and inefficient performance. Therefore, we propose a Text-driven Sparse Queries module to achieve sparse linguistic-aware query generation and facilitate a natural correspondence between queries and visual instances, paving the way for decoupling.

*4.3.1 The Generation Process.* In order to achieve a sparse distribution of initialized queries within the point cloud scene while preserving geometric and semantic information to a greater extent, we adopt the technique of farthest point sampling [52] directly for superpoints, namely the strategy of Farthest Superpoint Sampling (FSS):

$$P_i^S = \text{Pool}(K_i, P^P), \text{ FSS}(P^P) = \text{FPS}(P^S), \tag{9}$$

where $P^P \in \mathbb{R}^{N_P \times 3}$ denotes the positions of points in the original point cloud, $P^S \in \mathbb{R}^{N_S \times 3}$ denotes the positions of the superpoints, $K_i$ denotes the $i$-th superpoint's mask, Pool($\cdot$) is the operation of average pooling and FPS($\cdot$) is the operation of farthest point sampling [52], which returns the indices of the sampled points.

Subsequently, we utilize the features of superpoints obtained through Farthest superpoint sampling as initial seed queries, which participate in the subsequent stage of linguistic-aware refinement:

$$Q^{seed} = S[\text{FSS}(P^P)], P^{seed} = P^S[\text{FSS}(P^P)], \quad (10)$$

where $Q^{seed} \in \mathbb{R}^{N_{seed} \times D}$, $P^{seed} \in \mathbb{R}^{N_{seed} \times 3}$ denotes the features and the positions of seed queries, $N_{seed}$ denotes the total number of seed queries, $S$ denotes the features of superpoints and the $[\cdot]$ means the operation of indexing.

The $Q^{seed}$ obtained through farthest point sampling is linguistic-agnostic, tending towards comprehensive scene coverage [5, 43, 54]. However, the referring expressions often relate only to a subset of objects. Consequently, using $Q^{seed}$ directly as a query results in significant redundancy and interferes with the accurate determination of the target objects. Therefore, we further refined the selection of $Q^{seed}$ by ranking them based on their average correlation scores with the referring description features $T$, retaining the top $N_Q$ superpoints most relevant to the referring description:

$$R_i = \text{Average}(T \cdot Q_i^{seed}), \quad (11)$$

$$Q^f = Q^{seed}[\text{ArgTopk}(R, N_Q)], \quad (12)$$

where $R_i \in \mathbb{R}$ denotes the average correlation scores of the $i$-th seed query, $T \in \mathbb{R}^{N_T \times D}$ denotes the linguistic features, $Q_i^{seed}$ denotes the feature of the $i$-th seed query, $T \cdot Q_i^{seed}$ denotes the similarity matrix between $T$ and $Q_i^{seed}$, Average($\cdot$) denotes the averaging operation, $R \in \mathbb{R}^{N_{seed}}$ denotes the correlation scores of seed queries, ArgTopk($\cdot, N_Q$) returns the indices of the top $N_Q$ elements with the highest values and $Q^f \in \mathbb{R}^{N_Q \times D}$ denotes the finally selected superpoint features, which will serve as the queries.

*4.3.2 Query Generation Decoupling Loss.* To supervise the relevance scores of descriptions, a common approach is to employ the mentioned categories as labels for the seed queries. However, this simplistic approach often leads to queries that overly prioritize easily identifiable objects, resulting in the omission of other mentioned objects in the sampling process.

Therefore, we introduce Query Generation Decoupling Loss, which is categorized into three scenarios: Taking the $n$-th mentioned instance as an example, the label of the seed query closest to the center of the $n$-th mentioned instance is set to 1; for the remaining seed queries belonging to the $n$-th mentioned instance, their labels are determined using a Gaussian distribution based on their distances to the center of the $n$-th mentioned instance; the labels of seed queries not belonging to any instance are set to 0:

$$Q_n = \text{ArgTop1}(\text{Dist}(Q^{seed}, I_n^{mentioned})), \quad (13)$$

$$L_i^R = \begin{cases} 1 & \text{if} \quad Q_i^{seed} = Q_n^{seed}, \\ \exp(-\alpha \cdot \dfrac{dist_{i,n}^2}{\sigma^2}) & \text{if} \quad Q_i^{seed} \in I_n^{mentioned}, Q_i^{seed} \neq Q_n^{seed}, \\ 0 & \text{otherwise}, \end{cases}$$

$$(14)$$

where $Q_n$ denotes the seed query nearest to the center of the $n$-th mentioned instance, $I_n^{mentioned}$ denotes the $n$-th mentioned instance, $L_i^R$ denotes the relevance label of the $i$-th seed query, $Q_i$ denotes the $i$-th seed query, $dist_{i,n}$ denotes the distance from the $i$-th seed query to the center of the $n$-th mentioned instance, $\alpha$ is a control factor used to adjust the peak and shape of the Gaussian distribution, $\sigma$ is the standard deviation of the Gaussian distribution. Finally, the instance decoupling guidance loss is defined as:

$$\mathcal{L}_{qgd} = \text{BCE}(R, L^R), \quad (15)$$

where $R, L^R \in \mathbb{R}^{N_{seed}}$ denote the predictions and labels of the correlation scores for the seed queries.

## 4.4 Multi-object Decoupling Optimization

*4.4.1 Decoupling Mask Loss.* To refine the segmentation capability of each query, we continue to leverage the intrinsic attributes of queries generated by TSQ, where each query corresponds to an object within the point cloud. Building upon this foundation, we assign each query the responsibility of predicting the mask for its corresponding GroundTruth object. Specifically, for the $n$-th target instance GT mask $M_n^{tgt} \in \mathbb{R}^{N_S}$, we filter out the corresponding query $Q_n^+ \in \mathbb{R}^D$ and get the mask loss:

$$\mathcal{L}_{mask} = \text{BCE}(SW_M \cdot Q_n^+, M_n^{tgt}) + \text{DICE}(SW_M \cdot Q_n^+, M_n^{tgt}), \quad (16)$$

where $S \in \mathbb{R}^{N_S \times D}$ denotes the superpoint features, $W_M \in \mathbb{R}^{D \times D}$ is learnable parameters, BCE is the binary cross-entropy loss and DICE is the Dice loss [51]. $Q^+$ can be easily obtained by determining whether the query belongs to one of the target instances.

*4.4.2 Target Confidence Decoupling Loss.* For the $n$-th instance, we predicted the confidence score $A_n^{tgt}$ indicating the presence of the target object using the MDIN approach described in Section 4.2. Leveraging the generation method of our Text-driven Sparse Queries Generation, each query corresponds directly to the instance $I_{tgt}^n$, obtained through instance labels.

Consequently, we can construct labels $L^{tgt} \in \{0, 1\}^{N_Q}$ indicating whether each query contains the target object, where 1 represents the presence of the target and 0 represents its absence. Thus, we employ the binary cross-entropy (BCE) loss to enable each query to learn its discriminative ability regarding whether it belongs to the target object:

$$\mathcal{L}_{tgt} = \text{BCE}(A^{tgt}, L^{tgt}), \quad (17)$$

where $A^{tgt} \in \mathbb{R}^{N_Q}$ is the predicted confidence of queries and $L^{tgt} \in \{0, 1\}^{N_Q}$ denotes the corresponding label indicating whether the query contains one of the target instances.

**Table 1: Results of 3D-GRES task on Multi3DRes, where "zt w/ dis" means zero target with distractor, "zt w/o dis" means without distractor, "st w/ dis" means single target with distractor, "st w/o dis" means without distractor, "mt" means multiple target.**

| Method | mIoU | Acc@0.25 | | | | | | Acc@0.5 | | | | | |
|---|---|---|---|---|---|---|---|---|---|---|---|---|---|
| | | zt w/ dis | zt w/o dis | st w/ dis | st w/o dis | mt | Overall | zt w/ dis | zt w/o dis | st w/ dis | st w/o dis | mt | Overall |
| ReLA [35] | 42.8 | 36.2 | 72.7 | 48.3 | 83.4 | 73.0 | 61.8 | 36.2 | 72.7 | 20.4 | 65.5 | 42.4 | 37.4 |
| M3DRef-CLIP [80] | 37.4 | 39.2 | **81.6** | 50.8 | 77.5 | 66.8 | 55.7 | 39.2 | **81.6** | 29.4 | 67.4 | 41.0 | 37.5 |
| 3D-STMN [65] | 43.0 | 42.6 | 76.2 | 49.0 | 77.8 | 68.8 | 60.4 | 42.6 | 76.2 | 24.6 | 69.2 | 43.9 | 40.9 |
| Ours | 47.5 | 47.9 | 78.8 | **55.5** | 84.4 | 76.3 | 67.0 | 47.9 | 78.8 | **29.5** | 71.7 | 46.8 | 44.7 |

*4.4.3 Query-Text Alignment Loss.* To enhance the decoupling of queries and delegate them to individual instances, we leverage the intrinsic attributes of queries generated by TSQ. Each query originates from a superpoint within the point cloud, inherently associating it with a specific object. Queries for GroundTruth target instances are responsible for segmenting their corresponding instances, while unassociated instances are assigned to the nearest query. This method uses prior visual constraints to disentangle queries and assign them to individual instances.

Inspired by Wu et al. [69], we use the discriminative semantic features of referring text to align queries for GroundTruth instances with positive textual features $T^+$ (i.e., word features corresponding to target objects as described in Sec. 4.1.1), while pushing other queries away. This is achieved through a decoupled contrastive learning approach comprising query-word loss and word-query loss, defined as follows:

$$q = QW_q, \quad t = TW_w, \tag{18}$$

$$\mathcal{L}_{q \to w} = \sum_{i=1}^{N_Q} \frac{1}{|\mathbf{T}_i^+|} \sum_{t_i \in \mathbf{T}_i^+} - \log\left(\frac{\exp\left((q_i^\top t_i / \tau)\right)}{\sum_{j=1}^{N_T} \exp\left((q_i^\top t_j / \tau)\right)}\right), \tag{19}$$

$$\mathcal{L}_{w \to q} = \sum_{i=1}^{N_T} \frac{1}{|\mathbf{Q}_i^+|} \sum_{q_i \in \mathbf{Q}_i^+} - \log\left(\frac{\exp\left(t_i^\top q_i / \tau\right)}{\sum_{j=1}^{N_Q} \exp\left(t_i^\top q_j / \tau\right)}\right), \tag{20}$$

where $q \in \mathbb{R}^{N_Q \times C}, t \in \mathbb{R}^{N_T \times C}$ are the query and word features, $W_q, W_w \in \mathbb{R}^{D \times C}$ are learnable parameters. $N_Q$ and $N_T$ are the number of queries and words. $t_i \in T_i^+$ denotes the positive word feature of the $i$-th query and $q_i \in Q_i^+$ denotes the positive query feature corresponding to the $i$-th word. And we get the final Query-Text Alignment Loss $\mathcal{L}_{qta}$ by average $\mathcal{L}_{q \to w}$ and $\mathcal{L}_{w \to q}$:

$$\mathcal{L}_{qta} = \mathcal{L}_{q \to w} + \mathcal{L}_{w \to q}. \tag{21}$$

The final loss is calculated as the weighted sum of $\mathcal{L}_{qgd}$, $\mathcal{L}_{mask}$, $\mathcal{L}_{tgt}$ and $\mathcal{L}_{qta}$ :

$$\mathcal{L} = \lambda_{qgd}\mathcal{L}_{qgd} + \lambda_{mask}\mathcal{L}_{mask} + \lambda_{tgt}\mathcal{L}_{tgt} + \lambda_{qta}\mathcal{L}_{qta}, \tag{22}$$

where $\lambda_{qgd}$, $\lambda_{mask}$, $\lambda_{tgt}$ and $\lambda_{qta}$ are hyperparameters used to balance these losses.

## 5 Experiments

### 5.1 Experiment Settings

In our study, we utilize the pre-trained Sparse 3D U-Net [15] to extract point-wise features from the 3D point clouds and the pre-trained RoBERTa [42] as our text encoder. These pre-trained models serve as foundational components within our architecture. The remainder of the network is trained from scratch, starting with an

initial learning rate of 0.0001. To optimize this rate over the training period, we implement the PolyRL strategy, adjusting the learning rate with a decay power of 4.0. Our training procedure accommodates a batch size of 32 and processes text inputs with a maximum sentence length of 80 characters. To segment the original point cloud into manageable units, we adopt an unsupervised method for generating superpoints as outlined in [32, 60]. In TSQ, we first select a total of 256 seed queries using Farthest Superpoint Sampling. Subsequently, after filtering based on text correlation scores $R$, we end up with a final set of 128 queries, designated as $N_Q$. The MDIN is configured with six stacked layers to ensure robust feature integration and segmentation performance. We finely tune our network with specific loss weight settings: $\lambda_{qgd}$ is adjusted to 5, ensuring that query guidance loss is emphasized, while $\lambda_{mask}$ remains at 1. The weights for target and query-target alignment losses, $\lambda_{tgt}$ and $\lambda_{qta}$, are set more conservatively at 0.1 to balance the training dynamics. All experiments are conducted using the PyTorch framework on a single NVIDIA GeForce RTX 3090 GPU.

### 5.2 Quantitative Comparison

• **3D-GRES Results.** We present results of 3D-REC and RES models on the 3D-GRES task in Tab. 1, trained on the Multi3DRes benchmark. For single-stage networks like 3D-STMN [65], zero-targets are predicted for samples with fewer than 50 positive points [35]. Two-stage networks follow M3DRef-CLIP's [80] approach, classifying zero-targets if all instance scores are below $\tau_{pred}$.

Using ReLA [35] from 2D-GRES as our 3D baseline, we adapt it for 3D by initializing learnable queries similar to DETR [2], achieving an mIoU of 42.8%. M3DRef-CLIP performs slightly lower, and 3D-STMN, despite being state-of-the-art for 3D-RES, struggles with complex multi-object scenarios.

Our proposed model achieves the highest mIoU of 47.5%, surpassing M3DRef-CLIP by 10.1 points, and improves multiple target Acc@0.25 by 9.5 points, demonstrating strong discriminative capabilities for 3D-GRES.

• **Traditional 3D-RES Results.** As shown in Tab. 2, our proposed MDIN achieves substantial improvements over the previous state-of-the-art 3D-STMN [65] in traditional 3D-RES tasks. MDIN exhibits a 13.3-point increase in Acc@0.5 and an 8.8-point increase in mIoU. Notably, in scenes with multiple disruptive instances ("multiple"), MDIN demonstrates even greater enhancements, with a 10.3-point boost in mIoU and a 15.7-point increase in Acc@0.5. This indicates that our query decoupling approach not only improves traditional single-target 3D-RES tasks but also enhances semantic understanding and instance-level reasoning.

**Table 2: The traditional 3D-RES results on ScanRefer. † The mIoU and accuracy are reevaluated on our machine.**

| Method | Reference | Unique (~19%) 0.25 | 0.5 | Multiple (~81%) 0.25 | 0.5 | Overall 0.25 | 0.5 | mIoU |
|---|---|---|---|---|---|---|---|---|
| TGNN [26] | AAAI'21 | - | - | - | - | 37.5 | 31.4 | 27.8 |
| TGNN† [26] | AAAI'21 | 69.3 | 57.8 | 31.2 | 26.6 | 38.6 | 32.7 | 28.8 |
| InstanceRefer† [79] | ICCV'21 | 81.6 | 72.2 | 29.4 | 23.5 | 40.2 | 33.5 | 30.6 |
| X-RefSeg3D [55] | AAAI'24 | - | - | - | - | 40.3 | 33.8 | 29.9 |
| 3D-STMN [65] | AAAI'24 | 89.3 | 84.0 | 46.2 | 29.2 | 54.6 | 39.8 | 39.5 |
| SegPoint [18] | ACMMM'24 | - | - | - | - | - | - | 41.7 |
| RefMask3D [17] | ACMMM'24 | 89.6 | 84.7 | 48.1 | 40.8 | 55.9 | 49.2 | 44.9 |
| **MDIN (Ours)** | ACMMM'24 | **91.0** | **87.2** | **50.1** | **44.9** | **58.0** | **53.1** | **48.3** |

**Table 3: Ablation studies on designed modules, where "zt w/ dis" means zero target with distractor, "st w/ dis" means single target with distractor, "mt" means multiple target.**

| | TSQ | MOD | mIoU | Acc@0.25 Overall | Acc@0.5 zt w/ dis | st w/ dis | mt | Overall |
|---|---|---|---|---|---|---|---|---|
| 1 | × | × | 42.8 | 61.8 | 36.2 | 20.4 | 42.4 | 37.4 |
| 2 | ✓ | × | 44.4 | 63.2 | 42.5 | 24.6 | 42.3 | 40.5 |
| 3 | ✓ | ✓ | **47.5** | **67.0** | 47.9 | 29.5 | 46.8 | 44.7 |

**Table 4: Ablation studies on the components of the loss.**

| | $\mathcal{L}_{qgd}$ | $\mathcal{L}_{tgt}$ | $\mathcal{L}_{qta}$ | mIoU | Acc@0.25 Overall | Acc@0.5 zt w/ dis | st w/ dis | mt | Overall |
|---|---|---|---|---|---|---|---|---|---|
| 1 | × | × | × | 43.3 | 61.4 | 42.2 | 22.3 | 41.1 | 38.8 |
| 2 | × | ✓ | × | 45.2 | 65.5 | 45.8 | 24.6 | 43.1 | 41.1 |
| 3 | ✓ | ✓ | × | 46.4 | 65.5 | 46.8 | 26.3 | 43.8 | 42.4 |
| 4 | × | ✓ | ✓ | 46.2 | 65.9 | 46.5 | 26.2 | 45.3 | 42.3 |
| 5 | ✓ | ✓ | ✓ | **47.5** | **67.0** | 47.9 | 29.5 | 46.8 | 44.7 |

## 5.3 Ablation Study

*5.3.1 Ablation Study on Decoupling Modeling.* The decoupling in MDIN is achieved through TSQ and MDO. TSQ decouples queries in the physical space initially, while MDO refines them in the semantic space during optimization. We conducted an ablation study to assess their impact on 3D-GRES performance, as detailed in Tab. 3. The results indicate a significant performance drop when neither module is used. Incorporating TSQ improves the model's IoU by 1.6 points by refining queries and removing redundancy. Adding MDO further enhances performance, yielding a 4.5-point improvement in the challenging multiple-target Acc@0.5 metric, due to better query decoupling and semantic consistency.

*5.3.2 Ablation Study on Losses.* We conducted ablation studies on the loss components, as shown in Table 4. Rows 1 and 2 demonstrate that using $\mathcal{L}_{tgt}$ for query decoupling significantly boosts performance, with a 3.6-point increase in Acc@0.5 for zero targets with distractors. The inclusion of TSQ's $\mathcal{L}_{qgd}$, as seen in rows 2 and 3, and rows 4 and 5, substantially improves the Acc@0.25 metric for single targets, emphasizing the benefit of removing text-irrelevant redundancy for enhanced discriminability. Finally, $\mathcal{L}_{qta}$ notably increases the Acc@0.5 for multiple targets by 3 points (rows 3 and 5), due to its role in maintaining semantic consistency and selectively filtering queries aligned with specific semantic details.

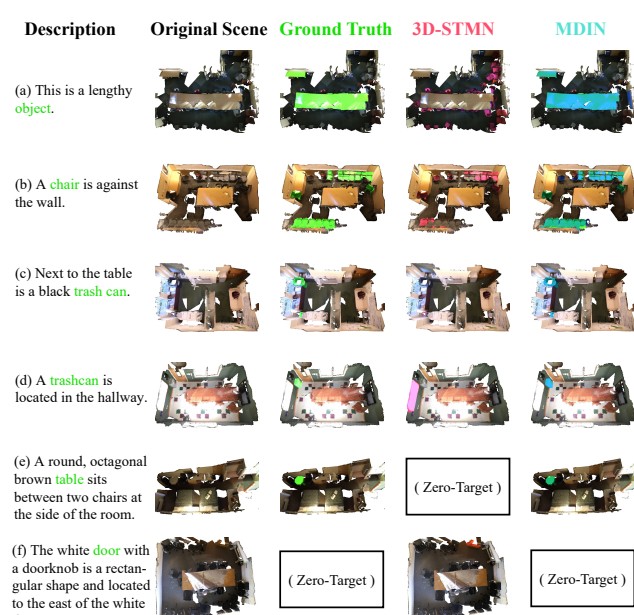

**Figure 3: Qualitative comparison between the proposed MDIN and 3D-STMN. Zoom in for the best view.**

## 6 Visualization

We visually compared the competitive 3D-STMN and MDIN models on the Multi3DRefer validation set, as shown in Fig. 3. MDIN excels in accurately segmenting challenging single-target cases and shows strong discriminative abilities for multiple and zero-target scenarios.

In instances with multiple objects, such as **(a)**, **(b)**, and **(c)** in Fig. 3, 3D-STMN struggles without decoupling, leading to semantic misidentification or missing targets. Conversely, MDIN effectively understands target semantics and segments all relevant objects. MDIN also performs well on single-target cases, as demonstrated in **(d)** and **(e)**. For extremely challenging non-target samples, like **(f)**, 3D-STMN's reliance on semantic feature similarity results in errors, while MDIN's decoupled approach correctly identifies and excludes non-target objects, making accurate zero-target predictions.

## 7 Conclusion

In this paper, we introduce 3D-GRES, a novel task for segmenting an arbitrary number of instances based on referential descriptions in 3D point clouds. We propose the Multi-Query Decoupled Interaction Network (MDIN) to address this task. MDIN features two key components: Text-driven Sparse Queries (TSQ) and Multi-object Decoupling Optimization (MDO). TSQ refines query generation by removing redundancies and retaining queries relevant to the textual context, while MDO enhances semantic consistency and query decoupling through various loss functions. Together, these components enable MDIN to set a new benchmark for performance on the 3D-GRES task, providing a robust and modular solution.

## Acknowledgments

This work was supported by National Key R&D Program of China (No. 2022ZD0118201), the National Science Fund for Distinguished Young Scholars (No. 62025603), the National Natural Science Foundation of China (No. U21B2037, No. U22B2051, No. 62072389, No. U21A20472), the National Natural Science Fund for Young Scholars of China (No. 62302411), China Postdoctoral Science Foundation (No. 2023M732948), the Natural Science Foundation of Fujian Province of China (No. 2021J01002, No. 2022J06001), and partially sponsored by CCF-NetEase ThunderFire Innovation Research Funding (NO. CCF-Netease 202301).

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
