# OpenReview forum: "3D-GRES: Generalized 3D Referring Expression Segmentation"
_acmmm.org/ACMMM/2024/Conference — MM2024 Oral_

### Official Review · Reviewer_CSu4 · 2024-05-07

**Rating:** 3
**Confidence:** 2

**Summary:**

In this paper, the author  introduce a new challenging 3D-GRES task and benchmark, exploring more general and flexible interaction in 3D
scenes through the development of the MDIN. Secondly, the author design the TSQ to ensure balanced coverage of key targets within the point cloud scene and the MDO strategy to maintain the distinctiveness of queries while preserving the semantic consistency of instructions.

**Strengths:**

This part of the work applies 2D GRES to the 3D field, which is innovative.

**Limitations:**

1. Insufficient workload in the experimental section.
2. Section 3.2 can be placed in the experiences section.
3. The author may consider validating the feasibility of the method on more other datasets.

**Suitability:**

3

---

### Official Review · Reviewer_MtWF · 2024-05-20

**Rating:** 5
**Confidence:** 3

**Summary:**

This paper presents a new and challenging 3D-GRES task, which focuses on segmenting any number of 3D instances based on natural language instructions. It outlines the task settings and introduces the Multi3DRes dataset, an enhancement of the Multi3DRefer dataset, to support the creation of a benchmark. The paper also proposes a baseline method featuring a novel multi-query decoupled interaction network to tackle this task. Experiments conducted on both 3D-GRES and traditional 3D-RES tasks demonstrate the effectiveness of the proposed method.

**Strengths:**

1. A baseline method called MDIN is proposed, and its effectiveness is verified through experiments. Specifically, a text-driven sparse queries generation module is introduced to create queries that ensure balanced coverage of key targets. Additionally, a multi-object decoupling optimization strategy is proposed to maintain the distinctiveness of the queries. Ablation studies demonstrate the effectiveness of these two fundamental components.
2. The benchmark is illustrated in detail, including task settings, dataset formulation, and metrics, making it accessible for future research.
3. The paper is well-written, with a clearly constructed and easy-to-follow method section.
4. Experiments comparing the proposed method to ReLA, M3DRef-CLIP, and 3D-STMN demonstrate its performance on this new task. The authors also include experiments on the classic 3D-RES task to showcase the state-of-the-art performance of the method. Ablation studies on the model components and hyperparameters validate the effectiveness of the proposed modules.

**Limitations:**

1. There is a typo on Line 347: 'projected linguistic feature' should be 'projected point feature.'
2. I suggest the authors describe in detail how superpoints are obtained and the operation of the point pooling layer.
3. I have a doubt about the appropriateness of using singular or plural forms in text descriptions, particularly in cases like the text description in Figure 3 (1): 'This is a lengthy object.' Here, the pronouns and nouns are in singular forms ('this', 'is', 'a', 'object'), but the segmentation target is a mask of multiple objects. Given that the vocabulary of pre-trained RoBERTa employs distinct word embeddings for singular and plural words (e.g., 'This' vs. 'These', 'object' vs. 'object + ##s'), could this difference in word embeddings impact the accurate extraction of text features?
4. Given that the authors introduce a novel benchmark comprising new task settings, datasets, and metrics, it raises the question of whether they plan to release their codebase for additional exploration.

**Suitability:**

3

---

### Official Review · Reviewer_68Ro · 2024-05-23

**Rating:** 5
**Confidence:** 3

**Summary:**

This paper proposes 3D-GRES, a new 3D referring expression segmentation task that segments objects within 3D space by a natural language query. To achieve this goal, the author proposes a new framework called the Multi-Query Decoupled Interaction Network (MDIN), which comprises two core components: Text-driven Sparse Queries (TSQ) and Multi-object Decoupling Optimization (MDO). Additionally, the author has constructed a new dataset, Multi3DRes, to accommodate this new task and conducted comprehensive evaluations, demonstrating significant improvements of this method over existing models.

**Strengths:**

1. The paper is well-organized and easy to follow.
2. The 3D-RES task is important as it can mine the semantic relationships between language and point cloud data, achieving multimodal 3D scene understanding. This paper innovatively extends segmenting individual objects to support more complex scene understanding tasks, broadening the limitations of the 3D-RES task and providing a finer-grained understanding of objects within a scene.

**Limitations:**

Although the manuscript proposes the new Multi3DRes dataset, it was actually built by re-annotating existing Multi3DRefer rather than using newly collected data. While this approach reduces the article's workload, it also limits its contribution.

**Suitability:**

3

---

### Meta-Review · Area_Chair_b8Jo · 2024-06-24

**Recommendation:** Accept (Oral)
**Confidence:** 5

**Metareview:**

This submission received two weak accept and one borderline accept ratings. As there is one missing final rating, the Area Chair reviewed the paper further and decided to accept the submission.